# Molecular Strategies to Diagnose Mucormycosis

**DOI:** 10.3390/jof5010024

**Published:** 2019-03-20

**Authors:** Laurence Millon, Emeline Scherer, Steffi Rocchi, Anne-Pauline Bellanger

**Affiliations:** 1Parasitology Mycology Department, University Hospital, 25000 Besancon, France; escherer@chu-besancon.fr (E.S.); Steffi.Rocchi@univ-fcomte.fr (S.R.); apbellanger@chu-besancon.fr (A.-P.B.); 2Chrono-Environnement UMR/CNRS 6249, University of Bourgogne Franche-Comté, 25000 Besançon, France

**Keywords:** mucormycosis, Mucorales, molecular diagnosis, PCR amplification, sequencing, real-time quantitative PCR

## Abstract

Molecular techniques have provided a new understanding of the epidemiology of mucormycosis and improved the diagnosis and therapeutic management of this life-threatening disease. PCR amplification and sequencing were first applied to better identify isolates that were grown from cultures of biopsies or bronchalveolar lavage samples that were collected in patients with Mucorales infection. Subsequently, molecular techniques were used to identify the fungus directly from the infected tissues or from bronchalveolar lavage, and they helped to accurately identify Mucorales fungi in tissue samples when the cultures were negative. However, these tools require invasive sampling (biospsy, bronchalveolar lavage), which is not feasible in patients in poor condition in Hematology or Intensive Care units. Very recently, PCR-based procedures to detect Mucorales DNA in non-invasive samples, such as plasma or serum, have proved successful in diagnosing mucormycosis early in all patients, whatever the clinical status, and these procedures are becoming essential to improving patient outcome.

## 1. Introduction

Murcormycosis refers to severe infectious diseases that are caused by filamentous fungi of the Mucorales order that primarily affect immunocompromised patients and patients with diabetes mellitus. An increasing incidence has been reported in Western countries, and mucormycosis has also been found in large numbers across India, especially in uncontrolled diabetics. This finding differs from that of developed countries, where the disease is more commonly diagnosed in patients with hematological malignancies and in transplant recipients [1,2]. An increase of about 7% per year was reported in the United States and France between 2000 and 2010, while the fatality rate increased by 9.3% per year [3,4]. In Spain, the disease incidence increased from 0.62 cases/100,000 admissions in 2005 to 3.3 cases/100 000 admissions during the 2007–2015 period [5]. 

Several factors may explain the growing incidence of mucormycosis, especially the increase in the number of susceptible people and the change in antifungal practices (particularly the prevention of invasive aspergillosis in high-risk groups). These factors may have altered the relative frequency of mucormycosis and aspergillosis among patients at risk for both infections [3]. In addition, the routine use of molecular techniques has helped in the accurate identification of Mucorales fungi in tissue samples when the cultures are negative. As a result, the number of mucormycosis cases that have diagnosed in the last 10 years has increased. 

First, molecular diagnosis was performed using the same samples as those that were used for cultures (tissue, bronchoalveolar lavage fluid (BAL)), which may be difficult to obtain in some patients, especially those in Hematology or Intensive Care units. Very recently, non-invasive PCR-based procedures have been used to detect Mucorales DNA in samples, such as plasma or serum [6,7,8,9,10] or even urine [11]. These techniques help to anticipate the diagnosis of mucormycosis and to distinguish between infections that are caused by Aspergillus and Mucorales very early. Moreover, they can be performed in every patient, whatever the clinical status, and they are becoming essential in improving patient outcome.

## 2. Molecular Techniques to Identify Mucorales from Cultures

Mucorales isolates that are grown on culture media can be identified by macroscopic and microscopic examination. The observation of broad, rare septate hyphae easily indicates fungi belonging to the Mucorales order. However, precise identification requires considerable mycological expertise and/or effective molecular tools. Mucorales may now be reliably identified using matrix-assisted laser desorption ionization–time of flight mass spectrometry (MALDI-TOF MS) techniques using either a commercial filamentous library (such as Bruker Daltonik for example) [12] or a house-made database [13].

Several molecular techniques targeting different loci (ribosomal targets 18S, 28S and Internal transcribed spacer (ITS); FTR1 gene, cytochrome b) were validated for the identification of Mucorales at the species level, including rare species and species that lack typical morphological characteristics [14,15,16]. Finally, ITS sequencing has been proposed as a valuable target for resolution to genus and usually to the species level by the CLSI guidelines for fungal identification [17]. 

Molecular identification of Mucorales species from culture is an interesting tool for *in vitro* diagnosis; however, cultures are often negative in patients with mucormycosis. Therefore, molecular techniques have also been used to directly identify the fungus from the tissue samples, especially when direct examination is positive.

## 3. Molecular Diagnosis in Tissue Samples

Two main approaches may be used to detect Mucorales in tissue samples.

The first approach consists of using panfungal primers, targeting ITS regions, followed by sequencing. This strategy was first validated in experimental models that were infected with Mucorales in both fresh and formalin-fixed paraffin-embedded (FFPE) samples [14,18], and it was then applied to clinical samples (both fresh and FFPE) [19,20]. This methodology proved to be very effective and it allowed for the successful identification of the causative Mucorales in most of the cases. The second possible approach consists of using Mucorales-specific primers. A semi-nested PCR targeting the 18S rDNA of Mucorales was first published in 2005 [21] and applied in several subsequent studies. On the whole, this semi-nested PCR specific for Mucorales detection was reliable to confirm tissue diagnosis and identify the causative agent in the case of a negative culture with a turnaround time of <48 h [22,23]. 

Finally, this semi-nested PCR specific for Mucorales detection was successfully modified into a real-time PCR format, followed by high-resolution melt analysis [24]. Diverse other Mucorales-specific PCRs were also tested: a multiplex real-time quantitative PCR (qPCR) targeting ITS1/ITS2 region with specific probes for *R. oryzae*, *R. microsporus*, and *Mucor* spp. [25], a real-time qPCR with specific primers that are designed to amplify a part of the cytochrome *b* gene [26], a specific qPCR targeting the 28S rDNA [27], two independent Mucorales specific real-time qPCR assays (targeting two different regions of the multicopy ribosomal operon-18S and 28S) that are able to detect DNA from a broad range of clinically relevant Mucorales species. [28].

In 2015, the contribution of PCR coupled with electrospray-ionization mass spectrometry (PCR/ESI-MS) was demonstrated for tissue samples with positive microscopy: although, the technique identified Mucorales to a species level very effectively and provided results within six hours; it is still expensive today (150 to 200 US $ per test) [29]. 

All of these different techniques were successful and confirmed that the PCR results were better in fresh/frozen samples than in FFPE samples, as highlighted by the European Society of Clinical Microbiology and Infectious Diseases (ESCMID) Fungal Infection Study Group (EFISG) and European Confederation of Medical Mycology (ECMM) joint clinical guidelines [30]. Indeed, whatever the techniques (PCR with panfungal primers or Mucorales specific primers, combination of specific quantitative PCR), the analytical sensitivity is about 56% to 80% when the FFPE samples are tested, and 97% to 100% when fresh tissue samples are tested [19,20,22,23,25,26,28].

## 4. Molecular Diagnosis in BAL Samples

Testing for Mucorales PCR on BAL fluid is an attractive approach that has been considered by several teams. In 2014, Lengerova et al. validated a PCR followed by a high-resolution melt analysis (PCR/HRMA) to detect Mucorales in BAL from the immunocompromised patients. The technique showed a high rate of sensitivity (100%) and specificity (93%), suggesting relevance for Mucorales DNA detection in BAL samples [31]. Recently, Springer et al. suggested using both the cell pellet and the supernatant of BAL to improve the sensitivity of the technique [32]. Scherer et al. used a combination of qPCR assays targeting 18S rDNA from *Mucor/Rhizopus*, *Lichtheimia,* and *Rhizomucor* previously described for detecting Mucorales DNA in serum [7] in order to test BAL specimens [33]. This study confirmed that Mucorales qPCR applied on BAL fluid could provide additional arguments favoring the earlier initiation of specific antifungal therapy, thus improving the outcome of pulmonary mucormycosis patients [33].

Although the effectiveness of BAL Mucorales qPCR testing has been well demonstrated, it is still difficult to obtain BAL sampling in patients who are in poor condition in Hematology or Intensive Care units.

## 5. Molecular Diagnosis on Blood Samples

Detecting Mucorales DNA with real-time qPCR in blood samples is now recognized as a non-invasive tool allowing for the early diagnosis of mucormycosis—as early as eight days before mycological diagnosis [6,7,8,34] and three days before imaging in patients with hematological malignancies [10].

qPCR detection of circulating DNA of Mucorales in serum or plasma can be prescribed as soon as possible after clinical suspicion of the diagnosis, and it may be performed in all patients, even those who cannot endure a biopsy or BAL. The qPCR techniques are fast (about 3h of turnaround time), specific, and cost-acceptable. For this reason, we and other research teams have suggested that serum/plasma Mucorales qPCR can be used to systematically screen high-risk patients with hematological malignancies [7,10] and severely burned patients [9]. 

The first technical option for performing Mucorales qPCR on blood samples was a combination of several genera-specific real time qPCR assays, which was developed to detect the most frequent genera involved in human diseases, based on local epidemiology (*Mucor/Rhizopus*, *Lichtheimia, Rhizomucor*) [6,7]. These assays showed sensitivity rates from 81% to 92%, depending on the volume of sample used for testing [7]. Recently, an additional genera-specific qPCR assay targeting *Cunninghamella* was developed [35]. Therefore, by optimizing the technical protocol with multiplex qPCR assays, the five most frequent genera can be detected in the same run, using three wells of a qPCR plate. 

Another technical option is a probe-based Mucorales-specific real-time qPCR assay that is able to detect DNA from a broad range of clinically relevant Mucorales species [8]. This technique, targeting specific fragments of the 18S rDNA gene, proved to be sensitive on serum samples and it can be performed while using only one well of a qPCR plate. However, an additional step of sequencing is necessary to identify the genera. In this assay, Mucorales DNA was detected in all sera from patients with probable/proven mucormycosis (100 %) and in 29 % of the possible cases of mucormycosis [8]. 

Although Mucorales DNA load that was found in serum of patients with mucormycosis is high (about 10 to 100-fold higher than *Aspergillus* DNA load found in serum of patients with invasive aspergillosis), the fact to use a large volume of serum or plasma (1 mL) is required to ensure an optimal sensitivity of qPCR blood-based techniques [7].

False positives with the Mucorales qPCR blood-based assays are very rare if stringent precautions against contamination are taken, but they may happen. The problem is often to differentiate false positive from very early diagnosis, especially when the result suggests a detection of very low DNA quantity (Cq > 41). In these cases, the close monitoring of the patient and a verification on a second serum (or plasma) as soon as possible are needed.

## 6. Molecular Diagnosis on Urine Samples

While investigating Mucorales and the host immunity response, a team discovered the gene family of spore coating encoding proteins (*CotH*) [36]. These *CotH* genes are only present in Mucorales. The fact that the *CotH* could be used as a target for early mucormycosis diagnosis was recently demonstrated while using animal models [11]. In this study, the PCR detection of CotH was positive for 3/3, 1/3, 2/2, and 3/3 urine samples from mice that were infected with *Rhizopus delemar*, *Lichtheimia corymbifera*, *Cunninghamella bertholettiae*, and *Mucor circinelloides*, respectively. PCR amplification of CotH was also positive in urine samples from four patients with proven mucormycosis. The authors conclude that the detection of *CotH* from urine samples was more reliable than from plasma or BAL fluid, with a sensitivity of 90% and a specificity of 100% for proven mucormycosis [11]. This result may be nuanced by the fact that heparinized plasma samples were tested, and thus PCR inhibition due to the presence of heparin was observed. Even so, the *CotH* genes showed great potential as universal biomarkers for mucormycosis infection and they should be validated on a larger number of patients.

## 7. Whole Genome Sequencing (WGS)/Mucormycosis Outbreak

Mucormycosis outbreaks have been internationally reported since 1977 [37]. Currently, the technique of whole genome sequencing (WGS) is available to investigate the link between the different strains that were isolated from patients when clustered cases occur, so that the epidemiology of the outbreak may be understood. WGS analysis was recently successfully applied to study an outbreak of invasive wound mucormycosis in a French hospital [38]. The outbreak was mainly due to multiple strains, and WGS was a useful tool in understanding the transmission patterns.

## 8. Highlight on the Pros and Cons for each Sample Type Used as Template for Molecular Diagnosis

The use of molecular techniques is now available in most of the primary care centers and cost is now acceptable. The turnaround technical time, including extraction and the Mucorales qPCR amplification run *per se*, is the same whatever the type of sample (BAL fluid, tissue biopsy, blood) and is relatively fast (about 3 h). However, the Mucorales qPCR is usually not performed every day at the lab, but once or twice per week. Some sample sites include more invasiveness (BAL fluid, tissue biopsy) and they present a higher sensitivity, because the sampling is close to the site of fungal growth. In these cases, direct examination may be positive before the Mucorales qPCR is done and the culture may also be positive. Mucorales qPCR performed on blood present the advantage of being totally noninvasive. Thus, this technique is appropriate for screening, especially in patients that cannot undergo BAL sampling or puncture. The sensitivity of blood-based Mucorales qPCR is inferior when compared to a Mucorales qPCR performed on tissue samples, but it is still largely acceptable. 

## 9. Benefit of More Extensive Use of Molecular Techniques

### 9.1. New Understanding of the Epidemiology of Mucormycosis

A recent meta-analysis indicated that molecular diagnosis has increased over the years (10% between 2000–2005 to 64% between 2011 and 2017) and it has provided new understanding of the epidemiology of mucormycosis [39]. Eight genera (28 species) were identified in 447/851 mucormycosis cases, of which *Rhizopus* was the most common (48%) followed by *Mucor* (14%), *Lichtheimia* (13%) *Cunninghamella* (7%), and *Rhizomucor* (6%). *Lichtheimia* infections were more frequent in Europe (29% in France [7,40] and 42% in Spain [5]; *Apophysomyces* infections were mainly reported in regions with tropical or warm climates, but not in Europe [39]. *Cunninghamella* spp. was more common in patients with pulmonary or disseminated disease, whereas *Rhizopus* spp. was more often isolated in patients with rhino-orbital cerebral mucormycosis. *Apophysomyses* and *Lichtheimia* were more often isolated in patients with cutaneaous mucormycosis. The overall mucormycosis mortality rate remained high (46%) and it varied according to genera (higher mortality among patients with *Cunninghamella* infections (77%) than in patients with *Lichtheimia, Rhizomucor, Mucor*, and *Rhizopus* infections (35%, 39%, 41%, and 47%, respectively) [39].

### 9.2. Improved Therapeutic Management

The use of molecular techniques has also improved the therapeutic management of mucormycosis. Indeed, mucormycosis and aspergillosis have similar underlying conditions and similar radiological and clinical signs, but their antifungal treatments are different. Mucorales infection requires amphotericin B lipid formulations, and it is not susceptible to voriconazole, the first line treatment of invasive aspergillosis. For each of these infections, early diagnosis and early initiation of directed treatment are essential in improving patient outcome. Molecular techniques make it possible to distinguish between the two infections, and to detect mixed Mucorales-*Aspergillus* infections. Moreover, the genera or species of Mucorales differ in their in vitro susceptibility, especially to posaconazole and isavuconazole, with the lowest MIC being observed for some *Rhizopus* species [41,42]. Molecular techniques may also provide fast, effective treatment by accurately identifying causative Mucorales genera.

In addition, the serum quantity of circulating DNA of Mucorales could be useful as a follow up tool to assess the fungal burden, which reflects the evolution of mucormycosis [43]. Accordingly, a negative serum qPCR was associated with a better patient outcome when compared to a qPCR that remained positive under antifungal treatment [7]. 

## 10. Conclusions

The molecular diagnosis of mucormycosis has improved the management of mucormycosis by the better identification of cultured isolates, but also by direct identification on clinical tissue samples. More recently, the early detection of circulating Mucorales DNA has been shown to be a major advance in the management of mucormycosis: the Mucorales qPCR performed on serum or plasma is a non-invasive technique that can be performed in all patients, with an acceptable time frame and a reasonable cost. The main problem at the moment is a lack of standardization, because of the use of multiple in-house assays. A commercial kit (Mucorgenius^®^, PathoNostics, Maastricht, The Nederlands) is now available, but the performance for clinical diagnosis needs to be evaluated. Efforts from the Fungal PCR Initiative (FPCRI) /Mucorales Lab working group of the International Society for Human and animal Mycology (ISHAM) are currently ongoing to improve standardization and provide recommendations, as was previously done for *Aspergillus* PCR assays by the European *Aspergillus* PCR Initiative (EAPCRI) working group [44].

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
