# Peer review of "Molecular Strategies to Diagnose Mucormycosis"

_jof, 2019, doi:10.3390/jof5010024_

Round 1

Reviewer 1 Report

The authors present a review about molecular diagnostic techniques for mucormycosis. As well outlined in the Introduction, this infection is becoming more prevalent with increasingly aggressive immunosuppressive strategies are used for many medical conditions.  However, standard culture-based diagnostic strategies are notoriously poor in this infection.

In this review, the authors do a very nice job summarizing the current literature about PCR-based assays to diagnose mucormycosis. There are a few points that, if addressed, would enhance the impact of this work.

1) The authors need to emphasize the inherent limitations of blood-based assays using PCR to capture microbial nucleic acids. While there would likely be sufficient organism burden in BAL fluid/cell pellets from a patient with pneumonia due to an invasive mold, there may not be adequate numbers of fungal in small aliquots of blood for an infection without a prominent blood stage. I did not appreciate any discussion of these limitations, only the suggestions of possibilities if the assays were sufficiently sensitive. Similarly, it would be helpful to consider potential causes of false positive results from molecular tests in this infection.

2) The authors mention MS-based diagnostic tests from clinical specimens, but no mention was given to using MS for organisms growing in culture.

3) I do not understand the meaning of the triangles in the Figure. Perhaps some enhanced text in the caption would help to make the point implied in this figure.

Author Response

We have carefully revised the manuscript. We took care to deal with each point raised by the editor and the reviewers. The changes are highlighted in yellow throughout the manuscript.

Reviewer 1

Comments and Suggestions for Authors

The authors present a review about molecular diagnostic techniques for mucormycosis. As well outlined in the Introduction, this infection is becoming more prevalent with increasingly aggressive immunosuppressive strategies are used for many medical conditions.  However, standard culture-based diagnostic strategies are notoriously poor in this infection. In this review, the authors do a very nice job summarizing the current literature about PCR-based assays to diagnose mucormycosis. There are a few points that, if addressed, would enhance the impact of this work.

Comment 1: The authors need to emphasize the inherent limitations of blood-based assays using PCR to capture microbial nucleic acids. While there would likely be sufficient organism burden in BAL fluid/cell pellets from a patient with pneumonia due to an invasive mold, there may not be adequate numbers of fungal in small aliquots of blood for an infection without a prominent blood stage. I did not appreciate any discussion of these limitations, only the suggestions of possibilities if the assays were sufficiently sensitive. Similarly, it would be helpful to consider potential causes of false positive results from molecular tests in this infection.

Author response:  

We agree that BAL represent a clinical sample originating from the site of infection, however procedures used to obtain BAL, and fractions (pellet/supernatant) used for DNA extraction, may influence the final DNA concentration in the volume tested (Springer 2018). In our experience, Cq value in BAL and in serum were not different  (median Cq, 32 cycles [range, 24 to 40 cycles] in BAL fluid versus 34 cycles [range, 31 to 41 cycles] in serum; P value = 0.14, Wilcoxon rank sum test for 17 paired samples). (Scherer 2018)

We also would like to highlight the high DNA load found in serum from patient with mucormycosis, which is very different from fungal DNA load found in patient with invasive aspergillosis, as previously reported by us and other teams:

Millon CMI 2016- page 810.e7   : “median Cq of 34 cycles (range 23-41 cycles)  corresponding to Mucorales DNA concentrations about 1-10 fg/mL of serum in patient with mucormycosis versus median Cq of 40 cycles ( range, 33-45 cycles) corresponding to Aspergillus DNA concentration of <0.1 fg/mL of serum in patients with invasive aspergillosis”

Springer J Med Microbiol 2016 – page 1419 “Similar Cq values for probable/proven IMM were recently reported (Millon et al., 2015). This reported Cq value (Cq 34) reflects a higher DNA load in patients with IMM than in patients affected by IA and may be caused by extensive angioinvasion by reflecting blood as a suitable diagnostic material.”

However, sensitivity is improved when using large volume of serum, as for Aspergillus PCR .

Therefore, page 8 lines 172-175 “Molecular diagnosis on blood samples” part, we added a paragraph on the importance of using a large volume of serum / plasma to ensure an optimal sensitivity of the blood-based PCR assays:

Although Mucorales DNA load found in serum of patients with mucormycosis is high (about 10 to 100 folds higher than Aspergillus DNA load found in serum of patients with invasive aspergillosis), the fact to use a large volume of serum or plasma (1 mL) is required to ensure an optimal sensitivity of qPCR blood-based techniques (Millon et al. 2016).

As recommended Page 8 lines 176-180 “Molecular diagnosis on blood samples” part, we also added a paragraph on the possibility of false positive results :

False positives with the Mucorales qPCR blood-based assays are very rare if stringent precautions against contamination are taken, but may happen. The problem is often to differentiate false positive from very early diagnosis, especially when the result suggests a detection of very low DNA quantity (Cq > 41). In these cases, close monitoring of the patient and a verification on a second serum (or plasma) as soon as possible are needed.

As also recommended by reviewer 2 , we added a paragraph that highlights the pros and cons for each sample type used as template for molecular diagnosis and highlight the higher sensitivity if sampling is close to site of fungal growth  (page 10- line 230-243)

Comment 2 The authors mention MS-based diagnostic tests from clinical specimens, but no mention was given to using MS for organisms growing in culture.

Author response:

As recommended Page 4 lines 73-76 “Molecular techniques to identify Mucorales from cultures” part, we added a paragraph on the possibility to identify reliably Mucorales using MALDI-TOF MS techniques “Mucorales may now be reliably identified using matrix-assisted laser desorption ionization–time of flight mass spectrometry (MALDI-TOF MS) techniques, using either a commercial filamentous library (such as Bruker Daltonik fro example) (Shao et al 2018) or a house-made database (Normand et al. 2017).

Comment 3: I do not understand the meaning of the triangles in the Figure. Perhaps some enhanced text in the caption would help to make the point implied in this figure.

Author response:

Triangles are used to classify the different techniques according to their invasiveness (“methods” triangle, from invasive to non invasive) and their speed to establish a diagnosis (“diagnosis” triangle from early to late). We added this information in the caption of the Figure.

Reviewer 2 Report

A review on Molecular strategies to diagnose Mucormycosis infection has to be considered a valuable work as it is indeed missing in the literature and at the present time there are many different techniques used in different labs.

The structure of the review is well thought and very concise, still trying to provide a comprehensive overview of the State of the Art.

That said, I find it a little misleading and not objective. There are several references missing, there is no mention of the delay in time for molecular techniques when applied to cultures or tissue samples, there is no mention for blood/serum samples about the negative aspects (not so effective in comparison to qPCR performed on tissue biopsies) and the paragraph on urine samples is not reporting the results of the study cited but only the first optimization.

Line 131 and 138: references are in a different style.

I would recommend to highlight the pros and cons for each sample type used as template for molecular diagnosis, elucidating also time, cost, invasiveness of the techniques and the potential of such techniques to be applied directly in primary care centers. Also I would suggest to be consistent and provide the results for sensitivity and specificity in percentage in each paragraph and not only in some of them, in order to offer a better and more objective comparison.

Author Response

We have carefully revised the manuscript. We took care to deal with each point raised by the editor and the reviewers. The changes are highlighted in yellow throughout the manuscript.

Reviewer 1

Comments and Suggestions for Authors

The authors present a review about molecular diagnostic techniques for mucormycosis. As well outlined in the Introduction, this infection is becoming more prevalent with increasingly aggressive immunosuppressive strategies are used for many medical conditions.  However, standard culture-based diagnostic strategies are notoriously poor in this infection. In this review, the authors do a very nice job summarizing the current literature about PCR-based assays to diagnose mucormycosis. There are a few points that, if addressed, would enhance the impact of this work.

Comment 1: The authors need to emphasize the inherent limitations of blood-based assays using PCR to capture microbial nucleic acids. While there would likely be sufficient organism burden in BAL fluid/cell pellets from a patient with pneumonia due to an invasive mold, there may not be adequate numbers of fungal in small aliquots of blood for an infection without a prominent blood stage. I did not appreciate any discussion of these limitations, only the suggestions of possibilities if the assays were sufficiently sensitive. Similarly, it would be helpful to consider potential causes of false positive results from molecular tests in this infection.

Author response:  

We agree that BAL represent a clinical sample originating from the site of infection, however procedures used to obtain BAL, and fractions (pellet/supernatant) used for DNA extraction, may influence the final DNA concentration in the volume tested (Springer 2018). In our experience, Cq value in BAL and in serum were not different  (median Cq, 32 cycles [range, 24 to 40 cycles] in BAL fluid versus 34 cycles [range, 31 to 41 cycles] in serum; P value = 0.14, Wilcoxon rank sum test for 17 paired samples). (Scherer 2018)

We also would like to highlight the high DNA load found in serum from patient with mucormycosis, which is very different from fungal DNA load found in patient with invasive aspergillosis, as previously reported by us and other teams:

Millon CMI 2016- page 810.e7   : “median Cq of 34 cycles (range 23-41 cycles)  corresponding to Mucorales DNA concentrations about 1-10 fg/mL of serum in patient with mucormycosis versus median Cq of 40 cycles ( range, 33-45 cycles) corresponding to Aspergillus DNA concentration of <0.1 fg/mL of serum in patients with invasive aspergillosis”

Springer J Med Microbiol 2016 – page 1419 “Similar Cq values for probable/proven IMM were recently reported (Millon et al., 2015). This reported Cq value (Cq 34) reflects a higher DNA load in patients with IMM than in patients affected by IA and may be caused by extensive angioinvasion by reflecting blood as a suitable diagnostic material.”

However, sensitivity is improved when using large volume of serum, as for Aspergillus PCR .

Therefore, page 8 lines 172-175 “Molecular diagnosis on blood samples” part, we added a paragraph on the importance of using a large volume of serum / plasma to ensure an optimal sensitivity of the blood-based PCR assays:

Although Mucorales DNA load found in serum of patients with mucormycosis is high (about 10 to 100 folds higher than Aspergillus DNA load found in serum of patients with invasive aspergillosis), the fact to use a large volume of serum or plasma (1 mL) is required to ensure an optimal sensitivity of qPCR blood-based techniques (Millon et al. 2016).

As recommended Page 8 lines 176-180 “Molecular diagnosis on blood samples” part, we also added a paragraph on the possibility of false positive results :

False positives with the Mucorales qPCR blood-based assays are very rare if stringent precautions against contamination are taken, but may happen. The problem is often to differentiate false positive from very early diagnosis, especially when the result suggests a detection of very low DNA quantity (Cq > 41). In these cases, close monitoring of the patient and a verification on a second serum (or plasma) as soon as possible are needed.

As also recommended by reviewer 2 , we added a paragraph that highlights the pros and cons for each sample type used as template for molecular diagnosis and highlight the higher sensitivity if sampling is close to site of fungal growth  (page 10- line 230-243)

Comment 2 The authors mention MS-based diagnostic tests from clinical specimens, but no mention was given to using MS for organisms growing in culture.

Author response:

As recommended Page 4 lines 73-76 “Molecular techniques to identify Mucorales from cultures” part, we added a paragraph on the possibility to identify reliably Mucorales using MALDI-TOF MS techniques “Mucorales may now be reliably identified using matrix-assisted laser desorption ionization–time of flight mass spectrometry (MALDI-TOF MS) techniques, using either a commercial filamentous library (such as Bruker Daltonik fro example) (Shao et al 2018) or a house-made database (Normand et al. 2017).

Comment 3: I do not understand the meaning of the triangles in the Figure. Perhaps some enhanced text in the caption would help to make the point implied in this figure.

Author response:

Triangles are used to classify the different techniques according to their invasiveness (“methods” triangle, from invasive to non invasive) and their speed to establish a diagnosis (“diagnosis” triangle from early to late). We added this information in the caption of the Figure.

Reviewer 2

Comments and Suggestions for Authors : A review on Molecular strategies to diagnose Mucormycosis infection has to be considered a valuable work as it is indeed missing in the literature and at the present time there are many different techniques used in different labs.

Comment 1: The structure of the review is well thought and very concise, still trying to provide a comprehensive overview of the State of the Art. That said, I find it a little misleading and not objective. There are several references missing, there is no mention of the delay in time for molecular techniques when applied to cultures or tissue samples, there is no mention for blood/serum samples about the negative aspects (not so effective in comparison to qPCR performed on tissue biopsies) and the paragraph on urine samples is not reporting the results of the study cited but only the first optimization.

Author response:

- 2 references were added (Normand, 2017; Shao 2018)

- The delay in time for molecular techniques have been indicated page 5- line 101, and page 10, line 235

-concerning the limitations of blood-based assays we added a paragraph about the need to  use large volume of serum or plasma for extraction to obtain optimal sensitivity,  page 8 lines 172-175 “Molecular diagnosis on blood samples” part:

Although Mucorales DNA load found in serum of patients with mucormycosis is high (about 10 to 100 folds higher than Aspergillus DNA load found in serum of patients with invasive aspergillosis), the fact to use a large volume of serum or plasma (1 mL) is required to ensure an optimal sensitivity of qPCR blood-based techniques (Millon et al. 2016).

- we also added a paragraph about the possibility of false positive results. Please see Page 8 lines 176-180 “Molecular diagnosis on blood samples” part:

False positives with the Mucorales qPCR blood-based assays are very rare if stringent precautions against contamination are taken, but may happen. The problem is often to differentiate false positive from very early diagnosis, especially when the result suggests a detection of very low DNA quantity (Cq > 41). In these cases, close monitoring of the patient and a verification on a second serum (or plasma) as soon as possible are needed.

-the part concerning the urine sample assays was rewritten in order to better present the results of the study  and not only the first optimization. Please see Page 8-9 lines 187-196

  In this study, the PCR detection of CotH was positive for 3/3, 1/3, 2/2, and 3/3 urine samples from mice infected with Rhizopus delemar, Lichtheimia corymbifera, Cunninghamella bertholettiae and Mucor circinelloides, respectively. PCR amplification of CotH was also positive in urine samples from 4 patients with proven mucormycosis. The authors conclude that detection of CotH from urine samples was more reliable than from plasma or BAL fluid, with a sensitivity of 90% and a specificity of 100% for proven mucormycosis (Baldin et al. 2018). This result may be nuanced by the fact that heparinized plasma samples were tested, and thus, PCR inhibition due to the presence of heparin was observed. Even so, CotH genes showed great potential as universal biomarkers for mucormycosis infection and should be validated on a larger number of patients.

Comment 2: Line 131 and 138: references are in a different style.

Author response:

All references were formatted using Endnote Web in the revised version

Comment 3: I would recommend to highlight the pros and cons for each sample type used as template for molecular diagnosis, elucidating also time, cost, invasiveness of the techniques and the potential of such techniques to be applied directly in primary care centers.

Author response:

as recommended we added a part Page 10 line 230-243 entitled “Highlight on the pros and cons for each sample type used as template for molecular diagnosis”

The use of molecular techniques is now available in most of the primary care centers and cost is now acceptable. The turnaround technical time including extraction and the Mucorales qPCR amplification run per se is the same whatever the type of sample (BAL fluid, tissue biopsy, blood) and relatively fast (about 3h). However, the Mucorales qPCR is usually not performed every day at the lab but once or twice per week. Some sample sites include more invasiveness (BAL fluid, tissue biopsy) and present a higher sensitivity because sampling is close to the site of fungal growth. In these cases, direct examination may be positive before the Mucorales qPCR is done, and the culture may also be positive. Mucorales qPCR performed on blood present the advantage of being totally noninvasive. Thus this technique is appropriate for screening, especially in patients that cannot undergo BAL sampling or puncture. The sensitivity of blood-based Mucorales qPCR is inferior compared to a Mucorales qPCR performed on tissue samples, but is still largely acceptable

Comment 4: Also I would suggest being consistent and providing the results for sensitivity and specificity in percentage in each paragraph and not only in some of them, in order to offer a better and more objective comparison.

Author response:

As recommended we checked for sensitivity and specificity in percentage for each study quoted (when found) and added the  in a concise manner, as follows:

- Molecular diagnosis in tissue samples, we added a paragraph page 6- lines 119-124 :

Indeed, whatever the techniques (PCR with panfungal primers or Mucorales specific primers, combination of specific quantitative PCR), analytical sensitivity is about 56 to 80% when FFPE samples are tested, and 97 to 100% when fresh tissue samples are tested (Buitrago et al. 2013; Lau et al. 2007; Hammond et al. 2011; Zaman et al. 2017 ; (Bernal-Martinez et al. 2013 ; Hata et al. 2008 ; Springer, Goldenberger, et al. 2016 ).

- Molecular diagnosis in BAL samples, the results for sensitivity / specificity were added pages 6, line 130

The technique showed a high rate of sensitivity (100%) and specificity (93%), suggesting relevance for Mucorales DNA detection in BAL samples (Lengerova et al. 2014)

Recently, Springer et al. suggested using both the cell pellet and the supernatant of BAL to improve the sensitivity of the technique (Springer et al. 2018).

-Molecular diagnosis on blood samples, the results for sensitivity / specificity were added page 7 lines 159-160,  and page 8 lines 169-171

The first technical option for performing Mucorales qPCR on blood samples was a combination of several genera-specific real time qPCR assays, developed  to detect the most frequent genera involved in human diseases, based on local epidemiology (Mucor/Rhizopus, Lichtheimia; Rhizomucor) (Millon et al. 2013, Millon et al. 2016). These assays showed sensitivity rates from 81% to 92%, depending on the volume of sample used for testing (Millon et al. 2016).

Another technical option is a probe-based Mucorales-specific real-time qPCR assay able to detect DNA from a broad range of clinically relevant Mucorales species (Springer, Lackner, et al. 2016). This technique, targeting specific fragments of the 18S rDNA gene, proved to be sensitive on serum samples, and can be performed using only one well of a qPCR plate. However, an additional step of sequencing is necessary to identify the genera. In this assay Mucorales DNA was detected in all sera from patients with probable/proven mucormycosis (100 %) and in 29 % of the possible cases of mucormycosis (Springer, Lackner, et al. 2016).

-Molecular diagnosis on urine samples , the data were added page 9 lines 190-192

The authors conclude that detection of  CotH from urine samples was more reliable than from plasma or BAL fuid, with a sensitivity of 90% and a specificity of 100% for proven mucormycosis (Baldin et al. 2018).
